# Epicardial Fat and Heart Failure in Type 2 Diabetes: Metabolism, Imaging and Novel Biomarkers—A Translational Perspective

**DOI:** 10.3390/jcm14238413

**Published:** 2025-11-27

**Authors:** Pedro Gil-Millan, José Rives, José Luis Sánchez-Quesada, Antonio Pérez

**Affiliations:** 1Endocrinology Department, Hospital Vall d’Hebron, 08035 Barcelona, Spain; pedroalejandro.gil@vallhebron.cat; 2Department of Medicine, Universitat Autònoma de Barcelona, 08193 Barcelona, Spain; 3Cardiovascular Biochemistry, Institut de Recerca Sant Pau (IR-Sant Pau), 08041 Barcelona, Spain; jrives@santpau.cat; 4Department of Biochemistry and Molecular Biology, Universitat Autònoma de Barcelona, 08193 Barcelona, Spain; 5Centro de Investigación Biomédica en Red of Diabetes and Metabolic Diseases (CIBERDEM), 28029 Madrid, Spain; 6Endocrinology Department, Hospital de la Santa Creu i Sant Pau, 08041 Barcelona, Spain

**Keywords:** epicardial adipose tissue (EAT), type 2 diabetes, heart failure with preserved ejection fraction (HFpEF), inflammatory biomarkers, sLRP1, GDF15, Galectin-3, sST2, LDL particle size, cardiac imaging

## Abstract

Heart failure (HF) is a major cardiovascular complication in people with type 2 diabetes (T2D), where heart failure with preserved ejection fraction (HFpEF) is the most common presentation. Despite its high prevalence, HF in T2D often remains undiagnosed during its early stages due to nonspecific symptoms and the limitations of conventional diagnostic tools. Epicardial adipose tissue (EAT), a visceral fat depot surrounding the myocardium, has emerged as a mechanistic and clinically relevant contributor to myocardial dysfunction. In T2D, EAT expansion fosters a pro-inflammatory, fibrotic, and metabolically adverse milieu that may directly promote the onset and progression of HF. This perspective synthesizes current translational evidence on the role of EAT in the pathogenesis of HF among individuals with T2D. We highlight diagnostic challenges related to imaging-based quantification and the limited sensitivity of natriuretic peptide-based screening, while emphasizing the potential relevance of emerging biomarkers such as GDF-15, Galectin-3, sST2, LDL particle size, GGT, and soluble low-density lipoprotein receptor-related protein 1 (sLRP1) to enhance early detection and risk stratification. Additionally, therapeutic approaches—including lifestyle modification, SGLT2 inhibitors, and GLP-1 receptor agonists—are considered for their potential to modulate EAT volume and reduce cardiovascular risk. Advancing knowledge on EAT biology and its circulating biomarkers holds promise to refine HF risk stratification and support translational efforts toward precision cardiometabolic care.

## 1. Introduction

### 1.1. Aim of the Review

This review adopts a translational narrative approach centered on epicardial adipose tissue (EAT) as a mechanistic driver of myocardial dysfunction in type 2 diabetes (T2D), with a special focus on the search of putative novel biomarkers of EAT volume. While EAT constitutes the core focus of the review, we also include systemic biomarkers that, although not exclusively produced by EAT, reflect complementary inflammatory, metabolic, or fibrotic pathways relevant to HFpEF. Throughout the manuscript, we explicitly differentiate EAT-derived biomarkers from systemic markers not directly originating from EAT. To guide the reader, the manuscript now progresses from global diabetic cardiomyopathy mechanisms to visceral adiposity, then to EAT biology, followed by diagnostic limitations and finally emerging biomarkers.

Methods of Review. We searched PubMed, Scopus, Embase, and Web of Science from 2000 to 2024 using terms related to ‘EAT’, ‘type 2 diabetes’, ‘HFpEF’, ‘cardiac imaging’, ‘adipokines’, ‘inflammatory and fibrotic biomarkers’, and LDL subfractions. We included mechanistic, imaging, translational, and clinical studies in adults with evidence levels ranging from 2 to 5. Case reports and non-original abstracts were excluded. Study heterogeneity precluded formal risk-of-bias scoring; observational limitations were considered when interpreting results.

### 1.2. Heart Failure in Type 2 Diabetes

Cardiovascular disease (CVD) remains the leading cause of death in individuals with type 2 diabetes (T2D). While coronary artery disease and stroke have traditionally dominated this landscape, heart failure (HF) has emerged as a highly prevalent yet frequently underdiagnosed complication, particularly heart failure with preserved ejection fraction (HFpEF) [1,2]. Epidemiological studies suggest that T2D confers a 2- to 5-fold higher risk of developing HF compared to non-diabetic individuals [3]. At the population level, the risk of HF in people with diabetes, including both type 1 diabetes (T1D) and T2D, reaches 74% above that of the general population [1,4].

The DIABET-IC study, a multicenter Spanish cohort, recently reported a 39.2% prevalence of HF among T2D outpatients, with HFpEF being the predominant phenotype [4]. More broadly, HF is recognized as a common complication of diabetes, with a prevalence of up to 22% and rising incidence rates. Incident HF was observed in 7.6% of patients over three years, particularly among older adults, those with chronic kidney disease, or with limited access to guideline-recommended therapies such as sodium–glucose cotransporter 2 inhibitors (SGLT2i) or glucagon-like peptide-1 receptor agonists (GLP-1 RAs) [4].

### 1.3. Diabetic Cardiomyopathy: A Distinct Clinical Entity

Diabetic Cardiomyopathy (DC) refers to diabetes-related myocardial dysfunction attributable to diabetes in the absence of other causes such as ischemic heart disease or hypertension [5]. Patients with T2D remain at increased risk of HF due to DC, which is characterized by structural and functional myocardial alterations driven by metabolic dysfunction [1,6]. The condition often begins with asymptomatic diastolic dysfunction and preserved ejection fraction, particularly in older adults, women, and individual with obesity [7]. Over time, it may progress to systolic dysfunction, increasing mortality and risk of hospitalization [8]. Factors accelerating DC include poor glycemic control, longer diabetes duration, and obesity [9,10]. In addition, albuminuria and end-stage kidney disease not only reflect microvascular burden but also independently heighten HF and cardiovascular mortality risk [11,12].

The pathogenesis of DC involves a complex interplay of molecular mechanisms which include metabolic disturbances, inflammation, oxidative stress, apoptosis, tissue remodeling and fibrosis (Figure 1). Insulin resistance impairs myocardial glucose uptake and promotes excessive uptake of circulating free fatty acids (FFAs) by cardiomyocytes, leading to intracellular lipid accumulation and myocardial lipotoxicity. This metabolic imbalance contributes to mitochondrial dysfunction, oxidative stress, endoplasmic reticulum (ER) stress and impairs myocardial contractility [13,14]. These putative mechanisms are summarized in Figure 1; it must be considered that most are conceptual and not proven causal pathways.

Hyperglycemia promotes advanced glycation end-products formation (AGEs) [15], which promote fibrosis, stiffening, and endothelial dysfunction through receptors for AGE (RAGE) signaling [16,17]. In parallel, hyperglycemia activates nuclear factor-κB-dependent cytokine pathways (TNF-α, IL-6, MCP-1), which further contribute to myocardial remodeling and fibrosis [6,18]. Excessive FFAs uptake by cardiomyocytes promotes lipotoxicity and oxidative stress, enhancing extracellular matrix (ECM) deposition by cardiac fibroblasts [19], and promoting cardiomyocyte hypertrophy and microvascular dysfunction [20]. Cardiac fibrosis is characterized by disproportionate deposition of collagen and dysregulation of matrix metalloproteinases (MMPs) and their tissue inhibitors (TIMPs) [21], leading to increased myocardial stiffness. Collectively, these processes culminate in diastolic dysfunction and contractile impairment, which constitute early hallmarks of DC. In addition, disruption of calcium handling further exacerbates contractile dysfunction and accelerates HF progression [22]. These metabolic and inflammatory disturbances converge on visceral adiposity, particularly epicardial adipose tissue (EAT), which serves as a key interface between diabetic metabolic dysfunction and myocardial injury.

### 1.4. Visceral Adiposity and Cardiac Dysfunction

Besides intrinsic myocardial alterations, T2D is associated with abnormal adipose tissue distribution, particularly an expansion of visceral adipose tissue (VAT) and ectopic depots such as epicardial adipose tissue (EAT). VAT is a major contributor to insulin resistance, systemic inflammation, and elevated cardiometabolic risk [23,24]. Ectopic fat accumulation in organs such as the liver or heart creates local lipotoxic and inflammatory environments that directly impair hepatocyte of cardiomyocyte function [25]. Among these depots, EAT has drawn increasing attention due to its anatomical proximity to the myocardium and coronary arteries. It may exert mechanical compression, secrete pro-inflammatory cytokines, and influence cardiac remodeling Via paracrine effects [26,27]. The next section describes the anatomical, physiological, and pathological features of EAT that contextualize its contribution to myocardial dysfunction in type 2 diabetes.

## 2. The Role of Epicardial Adipose Tissue in the Development of Heart Failure

### 2.1. Anatomy and Physiological Role

EAT is a visceral fat depot located between the myocardium and the visceral layer of the pericardium. Uniquely, it has no separating fascia from the underlying myocardium and shares the same coronary microcirculation [26,27]. Under physiological conditions, EAT acts as a mechanical cushion protecting the heart during contraction, provides thermoregulation, and functions primarily as a metabolic reservoir, supplying FFAs to meet myocardial energy demands, especially during increased workload [28,29].

### 2.2. EAT Expansion and Mechanical Constraint

In pathological conditions such as obesity and T2D, EAT volume can expand fourfold—from a typical 100 g in healthy individuals to more than 400 g [30]. This excessive expansion may restrict left ventricular (LV) compliance and contribute to diastolic dysfunction [28,31]. Observational studies have consistently reported associations between EAT volume and adverse cardiac remodeling, HFpEF, and coronary artery disease [32,33]. In post-myocardial infarction patients, EAT volume correlated with LV dysfunction, particularly in those with T2D [34]. While mechanical constraint may contribute, most evidence supports that the metabolic and paracrine activity of EAT plays a more decisive role than its physical burden [30].

### 2.3. Endocrine and Paracrine Activity of EAT

EAT is now recognized as an active endocrine organ, secreting adipokines and cytokines that influence myocardial and vascular biology (Figure 2). In healthy individuals, anti-inflammatory adipokines such as adiponectin contribute to vascular homeostasis. In contrast, in T2D and obesity, EAT undergoes adipocyte hypertrophy and inflammatory activation, with increased secretion of TNF-α, IL-6, MCP-1, and leptin [35,36]. These cytokines contribute to cardiomyocytes injury, apoptosis, and interstitial fibrosis [30]. Furthermore, inflammatory signaling from EAT also impairs coronary endothelium function, reducing nitric oxide bioavailability and vasodilatory capacity [28,37]. Therefore, it could also play a role in the accelerated development of coronary arteriosclerosis characteristic of individuals with diabetes or obesity.

Recent ex vivo studies further supporting the pathological role of EAT in T2D. Secretomes from diabetic patients, both with and without coronary artery disease, exhibited increased ceramide and saturated NEFA content, which induced inflammatory and cytotoxic responses in human cardiomyocytes. Interestingly, these effects were mitigated by apolipoprotein J and HDL, suggesting a potential therapeutic axis to buffer the adverse paracrine effects of diabetic EAT [38].

### 2.4. EAT as a Metabolic and Inflammatory Buffer

Besides mechanical and paracrine effects, EAT also modulates myocardial lipid metabolism by acting as a buffer for FFA uptake by the heart. In metabolic diseases such as diabetes or obesity, this buffering capacity is impaired, leading to lipid accumulation in cardiomyocytes and exacerbating insulin resistance [28]. EAT expresses numerous receptors that mediate FFA and glucose uptake, such as glucose transporter 4 (GLUT4), fatty acid binding protein 4 (FABP4), fatty acid transporter protein 4 (FATP4/CD36), retinol binding protein 4 (RBP4), and low-density lipoprotein receptor-related protein 1 (LRP1). The unbalanced expression of these receptors in EAT from diabetic patients and constitute potential targets of pleiotropic therapies for diabetes [39,40,41,42]. It must be noted that the endocrine/paracrine actions of EAT are based on associative and experimental evidence, not clinical trials. Therefore, direct causal validation is still missing.

## 3. Limitations in Traditional Diagnostic Approaches for Heart Failure in Type 2 Diabetes

Together, the commented mechanisms support a central role for EAT as an upstream modulator of myocardial stress; however, these early abnormalities often remain undetected by conventional diagnostic methods.

### 3.1. Plasma Biomarkers

Despite the mechanistic relevance of EAT, current diagnostic tools frequently fail to identify early EAT-related myocardial dysfunction. Diagnosing HF in individuals with T2D is particularly challenging in the early and asymptomatic stages. Observational data suggest that up to 30% of HFpEF cases in T2D patients are underdiagnosed [1,7], with onset frequently occurring within the first few years after diabetes diagnosis [6,43]. The current diagnostic framework relies predominantly on symptom-based evaluation, physical findings, and the exclusion of other etiologies. Circulating natriuretic peptides (NPs)—notably B-type natriuretic peptide (BNP) and N-terminal proBNP (NT-proBNP)—remain central tools [44]. However, their performance is limited in HFpEF and subclinical HF, particularly among patients with obesity. Adiposity suppresses NP synthesis and accelerates peptide clearance, yielding deceptively low levels despite early cardiac dysfunction [45]. Consequently, up to 20% of HFpEF patients may present “normal” NP values (BNP < 100 pg/mL; NT-proBNP < 125 pg/mL) [44,46,47,48]. Although atrial natriuretic peptide (ANP) levels may be elevated in T2D patients with diastolic dysfunction, available evidence is insufficient to support its diagnostic use in asymptomatic individuals [7]. Moreover, NPs offer limited ability to delineate disease stage, track disease progression, and enable risk stratification in the earliest phase. This diagnostic blind spot is compounded by the subtlety of HFpEF’s clinical manifestations and the confounding effects of obesity on both biomarker levels and imaging interpretation [2].

### 3.2. Imaging Analysis

Transthoracic echocardiography (EC) is the most widely used non-invasive imaging modality for assessing left ventricular (LV) structure and function, enabling detection of hypertrophy and diastolic dysfunction—core features of HFpEF [37]. A simple, standardized EC protocol should be as follows. The preferred anatomical window is the parasternal long-axis view, with optional short-axis confirmation. Measurements are obtained at end-systole, identifying the echo-free space between the myocardium and visceral pericardium, with calipers placed perpendicular to the right ventricular free wall. The mean of three cardiac cycles should be reported, and operators should complete a basic training set of ~30 supervised scans to ensure consistent landmark identification [37]. As an interim standard, reports should specify the window used, cardiac phase, caliper orientation, and cycles averaged. Based on published ranges and reproducibility, a ≥1 mm change may represent a meaningful variation, but this is hypothesis-generating, and establishing a validated MCID remains a priority for future consensus. Nonetheless, current guidelines do not recommend systematic screening in asymptomatic T2D patients, particularly those with recent diagnosis.

More advanced techniques, such as cardiac magnetic resonance imaging (MRI) and multi-detector computed tomography (MDCT), can quantify LV mass, fibrosis, and ischemia, and provide accurate assessments of EAT [31]. Yet these techniques are resource-intensive, lack standardized integration into routine pathways, and are therefore largely confined to research setting. Given the regulatory role of EAT on myocardial function and the observation that EAT expansion may precede overt functional impairment, quantifying EAT has potential to anticipate future HF. While MRI and MDCT offer precise volumetric measurements, echocardiographic EAT thickness provides an accessible, low-cost option suitable for clinical workflows [49,50,51]. However, the absence of universal thresholds, heterogeneous acquisition protocols, and uncertain incremental predictive value in asymptomatic populations have limited its adoption.

Collectively, these diagnostic limitations underscore a critical unmet need: scalable, sensitive tools capable of identifying early or subclinical HF in T2D. Emerging biomarker panels that capture key pathophysiological pathways such as inflammation, oxidative stress, and fibrosis, including markers linked to EAT dysfunction, may complement existing strategies. The next section examines these novel biomarkers, and their potential to enhance early HF detection and guide therapy.

## 4. Novel Biomarkers of Epicardial Fat and Heart Failure

For clarity, biomarkers are classified into (1) molecules directly secreted or dysregulated in EAT (adiponectin, leptin, IL-6, resistin, apelin, sLRP1), and (2) systemic biomarkers that reflect hepatic metabolism, systemic inflammation, fibrosis, or myocardial stress (sdLDL, GGT, Galectin-3, sST2, GDF-15). This distinction is maintained throughout each subsection.

Recent advances in cardiometabolic research have uncovered several biomarkers linked to EAT dysfunction and subclinical HF in T2D. Among these, inflammation-related markers, such as interleukin 6 (IL6), adiponectin, leptin, resistin, apelin, galectin 3, growth differentiation factor 15 (GDF15), or soluble suppression of tumorigenicity 2 (sST2) protein are emerging candidates. In addition, soluble low-density lipoprotein receptor-related protein 1 (sLRP1), small dense Low-Density Lipoprotein (sdLDL), and gamma-glutamyl transferase (GGT) have also been shown to be related with EAT volume. Unlike traditional markers such as natriuretic peptides (NPs) or troponins, these novel biomarkers reflect upstream inflammatory and metabolic processes and may offer novel insights into the earliest stages of diabetic cardiomyopathy.

### 4.1. Adiponectin and Leptin

Leptin and adiponectin, two key adipokines predominantly secreted by adipose tissue, exert opposing effects on metabolic homeostasis and inflammation: leptin promotes pro-inflammatory and metabolic activation, whereas adiponectin confers anti-inflammatory and insulin-sensitizing effects. Both molecules are secreted by EAT [50,51], where their functional balance may critically influence the local inflammatory microenvironment and cardiovascular health. Adiponectin improves endothelial function and reduces oxidative stress and inflammation, thereby protecting coronary circulation and cardiac function while leptin exhibits pro-inflammatory properties.

In the context of metabolic alterations, adiponectin secretion by EAT is reduced, leading to the loss of its protective functions and contributing to cardiovascular dysfunction. Several studies have shown that adiponectin expression or production by EAT is decreased in patients with coronary artery disease (CAD) [51,52,53,54,55], metabolic syndrome [56] or T2D [57]. In patients undergoing cardiac surgery, leptin expression in EAT was significantly elevated in those with CAD and was identified as an independent risk factor for adjacent coronary artery stenosis [58]. Leptin has been shown to promote oxidative stress and mitochondrial dysfunction in the myocardium, ultimately resulting in myocardial injury [59,60].

Serum adiponectin levels have been shown to negatively correlate with EAT volume in different populations. In a cross-sectional observational study [61], an inverse correlation was observed, indicating that lower adiponectin expression is associated with greater EAT accumulation. The same study reported a significant positive correlation between serum leptin levels and EAT area measured by computed tomography in cardiovascular surgery patients, also linking leptin with metabolic risk factors such as body mass index (BMI) and triglycerides.

### 4.2. Resistin and Apelin

Recent ex vivo research has identified resistin and apelin, cytokines secreted by macrophages in EAT, as potential biomarkers of adipose inflammation in CAD. In a case–control study comparing 21 CAD patients and 20 matched controls, CAD subjects exhibited significantly increased EAT resistin mRNA expression and decreased apelin mRNA, despite no differences in EAT volume or density across groups. These changes coincided with a pro-inflammatory macrophage phenotype—reflected in a higher CD11c/CD206 ratio—highlighting functional rather than volumetric EAT abnormalities [62].

Circulating resistin and apelin levels mirrored these tissue signatures, suggesting their promise as minimally invasive biomarkers of EAT dysfunction. Resistin activates the NF-κB signaling cascade, promoting IL-6 and TNF-α release and contributing to endothelial dysfunction and atherogenesis [63]. In contrast, apelin exerts cardioprotective and vasodilatory effects, improves insulin sensitivity, and counteracts resistin-induced cardiomyocyte hypertrophy Via AMPK/ERK pathways [64].

Clinically, reduced apelin has been linked with HF and adverse ventricular remodeling, while elevated resistin levels are associated with increased HF severity, hospitalization risk, and poor cardiovascular outcomes [63]. However, standardized assays are still lacking, and longitudinal data connecting resistin/apelin changes to EAT regression or HF progression are currently absent. Notably, in our recent study, despite increased levels in T2D patients, serum resistin did not correlate with indexed EAT volume (iEATv), suggesting it may reflect local inflammatory activity rather than total epicardial fat burden [65].

### 4.3. Interleukin 6

IL-6 is a pro-inflammatory cytokine that plays a central role in both systemic and local inflammation. IL-6 content in adipose tissue is up to 100-fold higher than in circulating plasma, suggesting it is an important regulator of adipose tissue inflammation [66]. Recent evidence has shown increased secretion of IL-6 by EAT in patients with CAD [67]. The secretion of IL-6 by EAT may contribute to a localized pro-inflammatory microenvironment which can directly impair myocardial function through paracrine interaction.

Elevated systemic IL-6 levels have been independently associated with reduced LVEF, atrial fibrillation and poorer clinical outcomes [68]. IL-6 has been linked also to obesity-related cardiac dysfunction, including impaired exercise capacity and elevated natriuretic peptides in HFpEF patients, even after adjusting for BMI [69]. Moreover, increased epicardial adipose tissue thickness has been positively associated with serum IL-6 levels and both EAT thickness and IL-6 were associated with arterial stiffness [70], supporting the hypothesis that EAT-driven inflammation may contribute to subclinical myocardial dysfunction. Additionally, IL-6 can interact with other adipokines and cytokines, amplifying tissue inflammation and sustaining the inflammatory loop between EAT and the surrounding tissues. These findings identify IL-6 not only as a biomarker of EAT but also as a potential therapeutic target in cardiac dysfunction driven by inflammation.

### 4.4. Soluble LRP1 (sLRP1)

LRP1 is a membrane receptor involved in postprandial lipid transport and insulin signaling, notably overexpressed in dysfunctional EAT. Its circulating form, sLRP1, serves as a non-invasive marker of adipose tissue volume [71,72,73]. A previous study has shown strong correlation between sLRP1 and EAT volume in type 1 diabetes patients [71]. Although an association analysis between EAT volume and plasma sLRP1 concentration has not been conducted in T2D, it has been reported that EAT from T2D patients show increased expression of LRP1. These findings suggest a potential role for sLRP1 in monitoring EAT-related cardiometabolic dysfunction.

### 4.5. Small Dense LDL and GGT

LDL particle size has gained attention as a marker of atherogenic dyslipidemia and insulin resistance [74]. Predominance of sdLDL particles (known as phenotype B of LDL) is associated with greater oxidative stress and higher EAT volume in T2D patients. In a previous study from our group, the size of LDL particles was independently associated with iEATv and markers of cardiometabolic risk [75]. On the other hand, GGT is a hepatic enzyme linked to liver dysfunction and cardiovascular risk [76]. In the same cohort, GGT was also associated with EAT expansion and other parameters of systemic inflammation. Other studies have shown that GGT plasma concentration is associated with VAT [77] and EAT thickness in patients with CAD [78]. Both LDL size and GGT, together with age, showed an additive capacity for predicting the volume of EAT in T2D. However, the size of this study was very small [75], and studies in larger cohorts are needed to establish these parameters as possible biomarkers of EAT volume. In addition, clinical adoption of sdLDL analysis is limited by cost and standardization challenges.

### 4.6. Galectin-3

Galectin-3 is a β-galactoside-binding lectin involved in a wide range of cellular processes, including inflammation, fibrogenesis, and tissue remodeling. Structurally, it consists of an N-terminal domain responsible for oligomerization and a C-terminal carbohydrate-recognition domain that enables interaction with glycosylated proteins [46]. It is primarily expressed by activated macrophages and fibroblasts and is secreted into the extracellular matrix in response to metabolic stress, where it orchestrates fibrotic remodeling through collagen cross-linking and fibroblast activation [46,79,80,81,82].

In the context of HFpEF, Galectin-3 levels are consistently elevated and correlate with disease severity, diastolic dysfunction, and long-term adverse outcomes including hospitalization and mortality [83,84]. This is particularly relevant in T2D, where Galectin-3 levels show strong associations with HbA1c, systemic inflammation (e.g., hsCRP) and LV mass, highlighting its potential as a mechanistic link between metabolic dysfunction and myocardial fibrosis [46,65,79,81,85,86]. Notably, Galectin-3 levels remain persistently elevated in newly diagnosed T2D patients even after intensive glycemic optimization and EAT volume reduction, suggesting ongoing inflammatory or fibrotic activity independent of glycemia [65].

The clinically accepted cut-off for Galectin-3 is >17.8 ng/mL, as demonstrated in the COACH study, where levels above this threshold were associated with increased mortality and rehospitalization in patients with chronic HF [87]. This finding was further supported by the HF-ACTION trial, which identified Galectin-3 as an independent predictor of mortality and cardiovascular events in individuals with HFrEF [88].

In our prospective T2D cohort, we identified that Galectin-3 was associated with elevated hsCRP and IL6 levels, reflecting low-grade inflammation even in early disease stages [65]. In addition, Galectin-3 was also associated with HbA1c and BMI [65]. While below the FDA-cleared threshold, this value of 17.8 ng/mL may represent a subclinical fibrotic phenotype in diabetic cardiomyopathy. This reinforces the possibility that earlier detection of Galectin-3 elevation in T2D could aid in identifying patients at higher cardiovascular risk before overt ventricular dysfunction develops.

### 4.7. Soluble ST2 (sST2)

sST2 is the soluble form of the interleukin-33 (IL-33) receptor and functions as a decoy that inhibits IL-33/ST2L-mediated cardioprotection [47,84]. In physiologic states, IL-33 binding to its transmembrane receptor ST2L on cardiomyocytes prevents apoptosis and fibrosis [89]. In pathological states such as T2D and HF, sST2 is upregulated in response to myocardial stretch, inflammation, and oxidative stress, thereby neutralizing IL-33 signaling and promoting adverse remodeling [84,89,90,91].

sST2 is a validated prognostic biomarker across the spectrum of HF phenotypes (HFpEF and HFrEF), and its levels independently predict cardiovascular death and rehospitalization [47,84,91]. In T2D, sST2 correlates with glycemic indices, triglycerides, and LV diastolic dysfunction severity, making it an attractive candidate for early detection of diabetic cardiomyopathy [65]. Uniquely, sST2 levels decline significantly after metabolic control and EAT reduction, indicating its utility as a dynamic biomarker of response to therapeutic intervention and adipose-driven myocardial stress [65]. In our study of newly diagnosed T2D patients [65], we identified that sST2 was independently associated with increased iEATv and markers of diastolic dysfunction.

Clinically, a cut-off value of ≥35 ng/mL has been widely adopted as a prognostic of HF and worse prognosis [92]. Elevated sST2 levels above this threshold are associated with a significantly increased risk of death and rehospitalization, independently of natriuretic peptides [93,94]. The PRIDE study was among the first to demonstrate that sST2 ≥ 35 ng/mL predicted 1-year mortality in patients presenting with dyspnea [93]. Subsequent validation has come from multiple cohorts, including where sST2 consistently improved risk stratification across acute and chronic HF populations [92,95,96].

While the value of 35 ng/mL is below the established high-risk threshold used in advanced HF, it may reflect earlier stages of diabetic cardiomyopathy, supporting the idea that lower sST2 cut-offs could enhance sensitivity in detecting subclinical HF in metabolically at-risk individuals. This underlines the need for population-specific validation and could support the integration of sST2 into personalized screening strategies for diabetic patients.

### 4.8. Growth Differentiation Factor 15 (GDF-15)

GDF-15, a stress-responsive cytokine of the TGF-β superfamily, plays a key role in mitochondrial homeostasis, cellular stress response, and systemic energy balance. It is secreted by cardiomyocytes under ischemic, oxidative, or metabolic stress and has emerged as a strong prognostic marker in both HF and chronic kidney disease [97,98]. Elevated levels of GDF-15 are linked to HF progression and mortality [47,84,99]. In T2D, GDF-15 predicts incident HF and is elevated in patients with worse insulin resistance, inflammation, or dialysis dependence [98,99].

While no single universally accepted clinical cut-off has been established, several studies have proposed risk stratification thresholds between 1200 and 2000 pg/mL [100]. In the PARADIGM-HF trial, patients with GDF-15 values in the top tertile had a two-fold higher risk of cardiovascular death [101]. Similarly, in the EMPEROR-preserved trial, GDF-15 was an independent predictor of all-cause mortality in patients with HFpEF, even after accounting for NT-proBNP [102]. The HF-ACTION substudy found that GDF-15 > 1700 pg/mL was associated with reduced VO2 peak and increased adverse events in HFrEF, suggesting that GDF-15 may reflect systemic stress and exercise intolerance [103] (see Table 1).

Our previous study demonstrated independent associations between indexed EAT volume and circulating GDF-15 levels in newly diagnosed T2D patients, reinforcing the role of visceral adipose tissue in modulating GDF15-driven stress pathways [65]. However, GDF-15 levels remained elevated despite 12 months of intensive metabolic control, suggesting that mitochondrial or adipose tissue dysfunction may persist beyond glycemic correction and continue to contribute to subclinical myocardial stress [65].

## 5. Integration of Plasma Biomarkers with EAT and Translational Evidence

Despite compelling evidence linking these biomarkers to HF and T2D, few studies have explored the relationship between potential upstream mediators of myocardial dysfunction and the increment of the volume of epicardial adipose tissue. In our recent prospective studies, we evaluated EAT volume and biomarker profiles in newly diagnosed T2D patients. In one of these studies, we found that GDF-15, sST2, and Galectin-3 levels were all elevated at baseline compared to healthy controls. iEATv was independently associated with GDF-15 and sST2, while Galectin-3 was associated with hsCRP. After 12 months of metabolic control, reductions were observed in iEATv and sST2, but not in Galectin-3 or GDF-15, indicating persistent inflammatory activity [65]. In the other study, we highlighted the existence of an axis that links liver function (GGT) with lipoprotein metabolism (LDL size) and the accumulation of iEAT, demonstrating that the upstream mechanisms that link these alterations are enormously complex and require a deep understanding of the metabolic pathways involved [75]. Therefore, these findings support the relevance of a multi-marker approach—anchored in EAT biology, but also in systemic inflammation, hepatic function and fibrotic factors—for early risk detection and therapy monitoring in diabetic heart disease. Table 1 summarizes the putative function and utility of biomarkers associated with EAT and HF in T2D described in the previous chapter.

Analytical and biological factors must be considered when interpreting biomarkers data. The methods used can yield disparate results depending on the platform employed for quantification. As summarized by Meijers et al. [47], sST2, Galectin-3, and GDF-15 are measured using immunoassays, including automated high-throughput platforms, but results from different manufacturers are not directly comparable, especially for sST2 where substantial inter-assay bias exists. Galectin-3 and GDF-15 show smaller but still relevant platform-dependent variability, meaning that cut-off values are assay-specific. Besides analytical variation, biomarker concentrations are influenced by pre-analytical factors, such as age, obesity, renal function, and systemic inflammation, which may independently elevate or suppress circulating levels. These confounders should therefore be considered when interpreting biomarker thresholds or planning reference ranges. Because robust, stratified reference intervals by BMI, age, and eGFR are not yet available, we identify areas where evidence is lacking and highlight the development of standardized, adjusted cut-offs as a priority for future consensus work.

In addition to these analytical and biological limitations, it is important to recognize that biomarker interpretation in HF is strongly modulated by the underlying clinical phenotype. HFpEF is not a uniform entity but encompasses distinct endotypes with specific comorbidity patterns, inflammatory profiles, and cardiometabolic stress signatures. Since EAT expansion, adipokine secretion, and inflammatory signaling differ across these endotypes, the expected biomarker response is also likely to vary. To illustrate this concept, and building on known HFpEF clusters, we propose a hypothesis-generating mapping of HFpEF endotypes to their anticipated EAT burden and dominant biomarker patterns (Table 2). This conceptual framework is intended to support translational thinking and guide future phenotyping efforts rather than define diagnostic thresholds.

## 6. Therapeutic Interventions: Impact on Heart Failure, Epicardial Adipose Tissue, and Biomarkers

As outlined earlier in this manuscript—namely, the persistent residual risk, early underdiagnosis, and the limited performance of traditional markers (especially in HFpEF and in early/subclinical disease)—there is a clear need to move beyond hemodynamic evaluation of HF in T2D. Going forward, the therapeutic management of HF in T2D should evolve from a purely hemodynamic paradigm toward strategies that integrate proven clinical benefits with targeted modulation of EAT and remodeling/inflammation biomarkers—particularly galectin-3, sST2, and GDF-15. This integrative perspective is likely to be most relevant in HFpEF phenotypes and in early or subclinical stages, where EAT burden and biomarker signals can refine early identification, anticipate progression, and guide treatment intensification and monitoring as thresholds and standardized algorithms are defined.

SGLT2 inhibitors (SGLT2i) consistently reduce HF hospitalizations across the HFrEF–HFpEF spectrum and are recommended in HFpEF according to EMPEROR-Preserved trial [104,105,106,107]. Beyond clinical benefit, human EAT expresses SGLT2; exposure to dapagliflozin increases glucose uptake and reduces pro-inflammatory cytokines in EAT explants, and in vivo studies show ~20% EAT volume reduction after 24 weeks, with similar results for canagliflozin [104,105,108]. In our cohort [65], with ~80% of patients on SGLT2i, 12 months of metabolic optimization were associated with a decrease in indexed EAT volume and in sST2, while Galectin-3 and GDF-15 remained elevated, suggesting persistent low-grade inflammation/stress despite glycemic and structural improvement [65]. Consistent with HF programs, GDF-15 may rise modestly under empagliflozin without indicating myocardial injury and retains independent prognostic value, potentially reflecting adaptive metabolic/stress responses [102].

GLP-1 receptor agonists (GLP1-RAs) show a modest reduction in HF hospitalization across CV outcome trials [44,109,110] and add a substrate-directed effect: EAT expresses GLP-1/GLP-2 receptors [111], and clinical studies report reductions in EAT thickness with liraglutide and dulaglutide, along with improved diastolic function with liraglutide versus placebo [112,113]. At the biomarker level, liraglutide lowers sST2 in obesity/prediabetes or early T2D, supporting an anti-fibrotic role [114]. In dual incretin therapy, tirzepatide—beyond its potent weight-loss effect—reduced left ventricular mass and paracardiac adipose tissue at 52 weeks in obesity-related HFpEF (SUMMIT CMR substudy), with concomitant improvements in LV and left-atrial structure, aligning structural benefits with weight loss [115]. Reductions in EAT and changes in sST2, Galectin-3, or GDF-15 with SGLT2i, GLP-1RA, or tirzepatide should be interpreted cautiously, as much of the observed effect—particularly with incretin-based therapies—is likely mediated by weight loss rather than direct adipose–cardiac actions. These signals remain mechanistic and hypothesis-generating and should not be equated with HF event reduction in the absence of dedicated outcome trials.

Weight-loss interventions—energy-restricted diets and structured exercise—improve diastolic filling and LV remodeling in obesity/early T2D [116,117], and bariatric surgery is associated with fewer HF hospitalizations in patients with established HF [118]. Diet and surgery both reduce EAT volume; the magnitude of EAT regression tracks BMI decline and is broadly similar across strategies for a given amount of weight loss [116]. In parallel, EAT regression is often accompanied by sST2 reductions that reflect favorable cardiometabolic remodeling, whereas persistent Galectin-3 may indicate residual fibrosis, and GDF-15 retains prognostic utility even when it changes with intervention [92,98,119,120,121,122,123]. These elements, summarized in Table 3, support serial multimarker follow-up to profile treatment response and residual risk.

Among other therapies with potential impact on EAT, metformin has been associated with modest reductions in EAT thickness, likely mediated by weight loss [124]; statins have shown decreases in EAT thickness in patients with coronary artery disease [125]; and DPP-4 inhibitors such as sitagliptin have demonstrated moderate EAT reductions correlated with VAT/BMI, although evidence for hard HF endpoints remains limited [126]. Thiazolidinediones reduce visceral adiposity but may increase HF risk, limiting their clinical use [31,127,128]. Overall, these mechanistic signals should be viewed as complementary to event-based evidence, and their translation to practice will require harmonized imaging protocols and universally accepted EAT thresholds (see Table 3).

Finally, combining therapies with proven event reduction (e.g., SGLT2i) with interventions that modify the adipose-myocardial substrate (incretin-based agents and structured weight loss), alongside multimarker monitoring—sST2 as a dynamic remodeling marker, Galectin-3 for residual fibrosis, and GDF-15 as a stress/adaptation index with prognostic value—offers a scalable pathway to optimize care in T2D at risk for HF. As EAT thresholds and standardization mature and biomarker-guided decision algorithms are validated, this approach should refine early stratification and more precisely guide treatment intensification [129].

**Table 3 jcm-14-08413-t003:** Therapeutic Interventions with Documented Effects on Epicardial Adipose Tissue in Type 2 Diabetes and/or Heart Failure.

Intervention	Effect on EAT	Effect on HF/Biomarkers
SGLT2 inhibitors (dapagliflozin, empagliflozin, canagliflozin)	↓ EAT volume by ~20% (dapagliflozin, 24 wks); ↓ inflammatory cytokines in EAT explants [104,105,108]	↓ sST2 after 12 months; Galectin-3 & GDF15 unchanged [65]; No effect reported sLRP1 [130] reduced HF hospitalization in HFpEF/HFrEF [106,107,131,132]
GLP-1 receptor agonists (liraglutide, dulaglutide, semaglutide)	↓ EAT thickness (liraglutide, dulaglutide); improved diastolic function [112,113,133]	Liraglutide ↓ sST2; Galectin-3 in obese/prediabetes; potential anti-fibrotic effect [114]
Dual GLP-1/GIP agonist (tirzepatide)	↓ Paracardiac adipose tissue by 45 mL. ↓ LV mass (52 wks) [134]	Improved LV/LA structure; reduced VAT inflammation in mice [115]
Weight loss (diet, exercise)	↓ EAT volume in meta-analysis; similar reduction with diet vs. bariatric surgery [43,116,117,118]	Improved diastolic filling and LV remodeling [43,116,117,118]
Metformin	↓ EAT thickness in newly diagnosed T2D [124]	Likely secondary to weight loss; unclear biomarker effects [124]
Statins	↓ EAT thickness in CAD [125]	Not well studied for HF biomarkers
DPP-4 inhibitors (sitagliptin)	↓ EAT thickness correlated with ↓ VAT and BMI [126]	Limited HF data
Bariatric surgery	↓ EAT volume significantly [118]	↓ HF hospitalization in obese HF patients [118]

Abbreviations: EAT, epicardial adipose tissue; HF, heart failure; VAT, visceral adipose tissue; LV, left ventricular; LA, left atrial; BMI, body mass index; T2D, type 2 diabetes; HFpEF, heart failure with preserved ejection fraction; HFrEF, heart failure with reduced ejection fraction. ↑ means increased; ↓ means decreased.

## 7. Conclusions

### Clinical Outlook and Future Directions

Taken together, current evidence supports a biologically coherent framework in which the adipose–myocardial axis—indexed by EAT and a panel of remodeling/inflammation biomarkers—contributes to the earliest stages of HF in T2D. In our incident T2D cohort, higher Galectin-3, sST2, and GDF-15 tracked with greater EAT burden, inflammatory milieu, and early ventricular remodeling, consistent with biomarker elevation preceding overt dysfunction in metabolically at-risk individuals. Additional signals related to hepatic stress (e.g., GGT), lipoprotein quality/function (e.g., LDL particle size, sLRP1), and adipokine-inflammation pathways (adiponectin, leptin, resistin, apelin, IL-6) further suggest a multisystem fingerprint that may help anticipate EAT expansion and downstream HF risk in T2D. Figure 3 illustrates a hypothetical framework in which EAT-related imaging findings are complemented by circulating biomarkers (sST2, Galectin-3, GDF-15, or others) to refine early risk stratification. This proposed combination of imaging and biomarker profiling is exploratory and requires validation in prospective, adequately powered T2D cohorts before clinical implementation.

Translating these insights into routine care will require prospective, adequately powered, and methodologically harmonized studies. Priorities include (i) standardizing pre-analytical and analytical conditions for biomarkers and defining population-specific cut-offs with clear clinical decision points; (ii) harmonizing EAT assessment across echocardiography and RMI/CT (measurement planes, acquisition parameters, and reporting) and converging on universally interpretable thresholds; and (iii) proving incremental value beyond natriuretic peptides and conventional imaging—ideally with patient-centered outcomes, cost-effectiveness, and equity analyses. A practical pathway for primary care should include who to test (age, duration of T2D, BMI), which panel (sST2, Gal-3, GDF-15, ….), when to repeat, and when to refer for imaging. Figure 4 shows a hypothetical pathway and requires validation.

As a provisional research framework, we propose a functional biomarker hierarchy in which sST2 reflects dynamic myocardial stress and remodeling, Galectin-3 denotes residual fibrotic activity, and GDF-15 serves as a systemic stress and adaptation index. This hierarchy is exploratory and hypothesis-generating, and its clinical utility must be validated in prospective T2D cohorts. Future studies should assess incremental prognostic value using decision-curve analysis and reclassification metrics such as the Net Reclassification Improvement (NRI) and the Integrated Discrimination Improvement (IDI). These analyses will be essential to determine whether this hierarchy meaningfully enhances HFpEF risk stratification.

Looking ahead, multimarker strategies embedded in clinical pathways—and augmented by transparent AI/ML models—could enable risk-tailored screening, earlier initiation of disease-modifying therapy, and targeted follow-up in high-risk T2D. On the therapeutic side, SGLT2 inhibitors provide an event-reducing foundation, while incretin-based therapies and structured weight loss appear to remodel adiposity (including paracardiac depots) and may shift biomarker trajectories. Whether treat-to-biomarker approaches (e.g., sST2-guided intensification, Gal-3-focused antifibrotic strategies, nuanced interpretation of GDF-15 as a stress/adaptation index) translate into fewer HF events remains a critical question for practical trials.

## Figures and Tables

**Figure 1 jcm-14-08413-f001:**
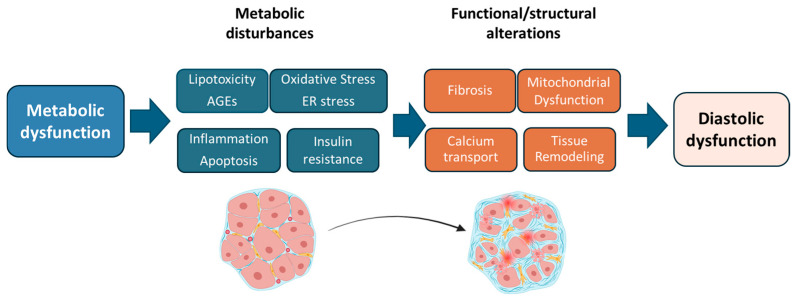
In T2D, systemic metabolic dysfunction triggers structural and functional alterations in the myocardium that result in asymptomatic diastolic dysfunction and HFpEF. Diabetic cardiomyopathy (DC) involves an interplay of metabolic disturbances in the myocardium, including lipotoxicity, advanced glycation end-products (AGEs), inflammation, apoptosis, oxidative stress, endoplasmic reticulum (ER) stress, and insulin resistance. These processes lead to tissue fibrosis and remodeling, mitochondrial dysfunction, and impaired calcium handling, ultimately causing diastolic dysfunction and contractile impairment that promote DC development and progression to HF.

**Figure 2 jcm-14-08413-f002:**
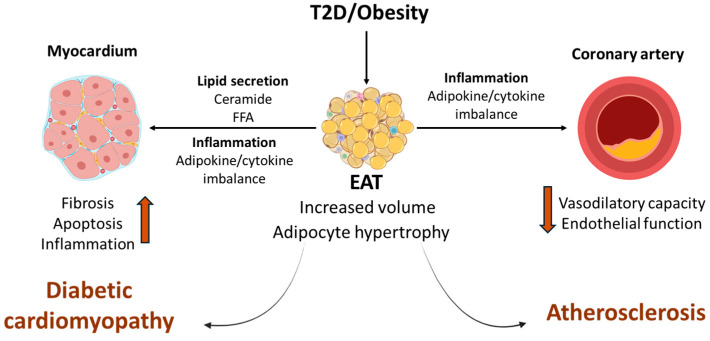
In T2D and obesity, EAT increases in volume and undergoes adipocyte hypertrophy and inflammatory activation, characterized by increased secretion of pro-inflammatory cytokines and decreased secretion of anti-inflammatory cytokines. In diabetic patients, EAT also exhibits enhanced lipid secretion leading to lipid accumulation and lipotoxicity. Together, these inflammatory and lipotoxic effects promote myocardial fibrosis, inflammation, and apoptosis, which ultimately lead to DC. Increased inflammatory signaling in EAT also impairs endothelial function and reduces vasodilatory capacity in the coronary arteries, thereby mediating the development of atherosclerosis.

**Figure 3 jcm-14-08413-f003:**
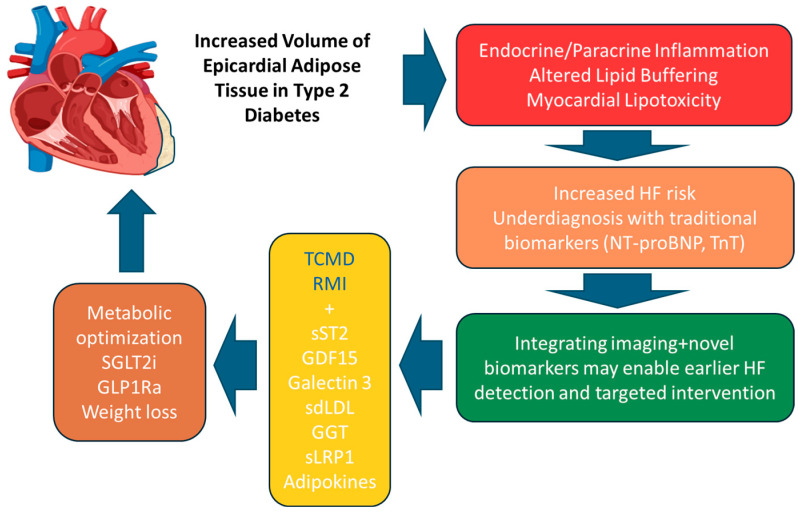
The inclusion of new markers that can complement imaging analyses to determine EAT volume can be an extremely useful tool for cardiovascular risk stratification and facilitate the implementation of appropriate therapeutic strategies.

**Figure 4 jcm-14-08413-f004:**
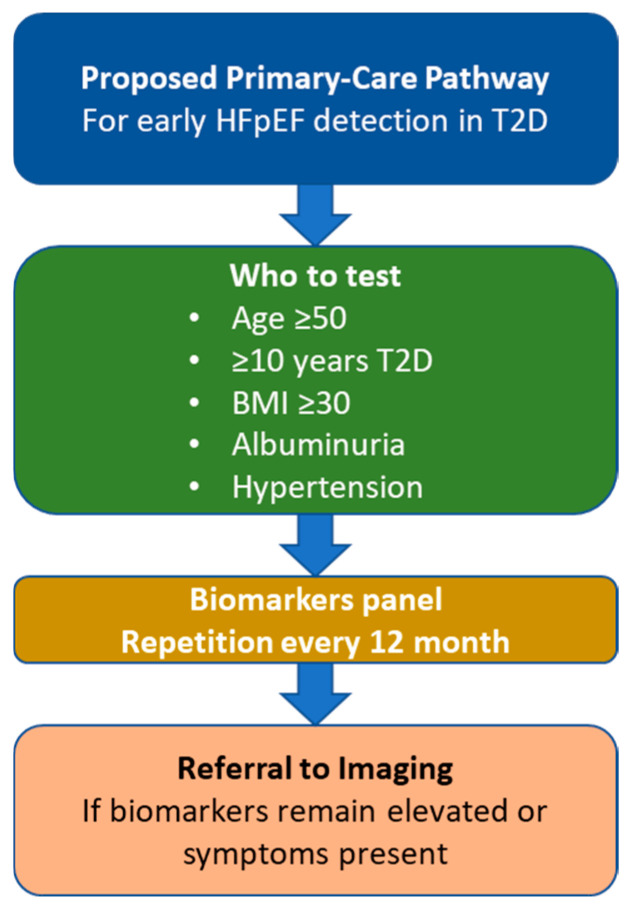
Proposed Primary-Care Pathway for Early HFpEF Detection in Type 2 Diabetes.

**Table 1 jcm-14-08413-t001:** Biomarkers Associated with Epicardial Adipose Tissue and Heart Failure in Type 2 Diabetes.

Biomarker	Function/Mechanism/Pathway	Assay Method/Platform ^1^	Reference Cut-Off (If Available) ^2^	Association with iEATv	Clinical/Research Applicability
Galectin-3	β-galactoside-binding lectin; fibrosis, ECM remodeling, inflammation	Microtiter-plate-based sandwich ELISA (BG Medicine, Inc.)	>17.8 ng/mL [87]	No independent association after adjustment; correlated with hsCRP [65,80]	Early fibrotic phenotype detection; prognostic; therapeutic target
sST2	IL-33 decoy receptor; myocardial stress, inflammation	Quantitative sandwich monoclonal ELISA (Critical Diagnostics, Inc.)	≥35 ng/mL [95]	Independent positive association [65]	Prognostic marker; therapeutic monitoring; early myocardial strain
GDF-15	TGF-β superfamily; mitochondrial stress, oxidative stress, metabolic regulation	Automated quantitative sandwich immunoassay (Roche Diagnostics)	>1500–2000 pg/mL [103]	Independent positive association [65]	Predicts HF incidence; metabolic modulation target
sLRP1	Lipid uptake regulator; adipose overexpression	Sandwich ELISA	N/A	Positive correlation with EAT [73]	EAT activity marker; lipid metabolism modulation
LDL size (sdLDL)	Atherogenic dyslipidemia; oxidative stress	Gradient gel electrophoresis	N/A	Positive association [75]	Atherogenic risk; limited by assay availability
GGT	Oxidative stress, hepatic function	Dry chemistry, automated enzymatic assay	N/A	Positive association [77,78]	Liver–EAT–CVD link; oxidative stress marker
IL-6	Pro-inflammatory cytokine; vascular dysfunction	ELISA and Multiplexed immunoassay	N/A	Positive association with EAT thickness; correlated with arterial stiffness [66,67,68,69,70]	Inflammation marker; therapeutic target
Adiponectin	Anti-inflammatory adipokine	ELISA and Multiplexed immunoassay	N/A	Inverse correlation with EAT volume [50,51,52,53,54,55,56,57,61]	Protective; decrease linked to EAT dysfunction
Leptin	Pro-inflammatory adipokine	ELISA and Multiplexed immunoassay	N/A	Positive correlation with EAT volume [50,58,60,61]	Obesity and EAT-related inflammation
Resistin	Macrophage-derived adipokine; inflammation	ELISA and Multiplexed immunoassay	N/A	No correlation with EAT [65]	Local inflammatory activity marker
Apelin	Cardioprotective adipokine	ELISA	N/A	Inverse association with EAT inflammation [62,64]	Vasodilatory & metabolic benefits

Abbreviations: EAT, epicardial adipose tissue; ECM, extracellular matrix; ELISA, enzyme-linked immunosorbent assay; HF, heart failure; hsCRP, high-sensitivity C-reactive protein; iEATv, indexed epicardial adipose tissue volume; IL, interleukin; N/A, not available; sLRP1, soluble low-density lipoprotein receptor-related protein 1; sST2, soluble suppression of tumorigenicity-2; sdLDL, small dense low-density lipoprotein; GGT, gamma-glutamyl transferase; ^1^ When the trademark is indicated, it refers to the reference where the cut-off is described. Other immunoassays may be available under different brands. ^2^ Reference cut-offs for HF-specific risk are only available for sST2, galectin-3 and GDF-15. These cut-offs are specific for these assays.

**Table 2 jcm-14-08413-t002:** Conceptual Integration of HFpEF Endotypes with EAT Burden and Biomarker Patterns.

HFpEF Endotype	Expected EAT Burden	Dominant Biomarkers	Pathophysiological Note
Obese/Cardiometabolic	High	↑ IL-6, ↑ leptin, ↑ sST2; ↑ GDF15	EAT-driven inflammation and metabolic stress
Hypertensive/Aging	Moderate	↑ Galectin-3; ↑ GDF15	Fibrotic and structural remodeling phenotype
Atrial fibrillation/Atrial myopathy	Variable	↑ GDF-15	Systemic stress and atrial remodeling
Pulmonary vascular/Right-sided HFpEF	Low–moderate	↑ sST2, ↑ GDF-15	Cardiopulmonary strain and RV load

Abbreviations: EAT, epicardial adipose tissue; GDF15, growth differentiation factor 15; HFpEF, HF with preserved ejection fraction; sST2, soluble suppression of tumorigenicity-2. ↑ means increased concentration.

## Data Availability

No new data were created or analyzed in this study. Data sharing is not applicable to this article.

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
