# Peer review of "J. Clin. Med.2025, 14(23), 8413;https://doi.org/10.3390/jcm14238413"

_jcm, 2025, doi:10.3390/jcm14238413_

Round 1

Reviewer 1 Report

Comments and Suggestions for Authors

The title of the article Epicardial Fat and Heart Failure in Type 2 Diabetes: Metabolic
Modulation, Imaging and Inflammatory Biomarkers—A Translational Perspective
implies a focus on the role of epicardial adipose tissue (EAT) in relation to diabetic HFpEF. At the same time, the main focus of the article presented is on the pathogenic features and biomarkers of diabetic cardiomyopathy that can lead to HFpEF. The authors discuss various biomarkers in the context of HFpEF that have both direct and indirect connections with EAT. For example, galectin-3, sST2 and GDF-15 are discussed in the manuscript in the context of its novelty, but they are not directly related to EAT. The authors address the connection between lowering the level of pro-inflammatory and pro-fibrotic molecules and the thickness of EAT through therapeutic interventions. However, it is important to note that EAT is not the only source of pro-inflammatory cytokines and other «biomarkers» related to HF, and reducing their concentration in blood does not necessarily mean a reduction in production specifically from EAT. Consequently, the authors should more clearly indicate which molecules are produced directly by EAT and which are not directly related to it. Due to the lack of logical connection between the sections, it is not entirely clear what task the authors set themselves when writing this review. My recommendation is that the authors should change the title and/or the order of presentation of information in their manuscript.

Author Response

Reviewer 1

The title of the article Epicardial Fat and Heart Failure in Type 2 Diabetes: Metabolic

Modulation, Imaging and Inflammatory Biomarkers—A Translational Perspective implies a focus on the role of epicardial adipose tissue (EAT) in relation to diabetic HFpEF. At the same time, the main focus of the article presented is on the pathogenic features and biomarkers of diabetic cardiomyopathy that can lead to HFpEF. The authors discuss various biomarkers in the context of HFpEF that have both direct and indirect connections with EAT. For example, galectin-3, sST2 and GDF-15 are discussed in the manuscript in the context of its novelty, but they are not directly related to EAT. The authors address the connection between lowering the level of pro-inflammatory and pro-fibrotic molecules and the thickness of EAT through therapeutic interventions. However, it is important to note that EAT is not the only source of pro-inflammatory cytokines and other «biomarkers» related to HF, and reducing their concentration in blood does not necessarily mean a reduction in production specifically from EAT. Consequently, the authors should more clearly indicate which molecules are produced directly by EAT and which are not directly related to it. Due to the lack of logical connection between the sections, it is not entirely clear what task the authors set themselves when writing this review. My recommendation is that the authors should change the title and/or the order of presentation of information in their manuscript.

We sincerely thank the reviewer for these insightful comments, which have substantially improved the clarity, coherence, and focus of the manuscript. Following the reviewer’s recommendations, we implemented several major revisions.

First, we have clarified throughout the text which biomarkers are directly produced by, or strongly modulated by, epicardial adipose tissue (EAT)—including adiponectin, leptin, IL-6 from EAT, resistin, apelin, and sLRP1—versus systemic biomarkers that mainly reflect hepatic metabolism, systemic inflammation, fibrosis, or myocardial stress (LDL particle size, GGT, Galectin-3, sST2, GDF-15). These distinctions are now explicitly stated in the Introduction and in each biomarker subsection to prevent any ambiguity regarding their origin. Page 7, lines 251-255

Second, we improved the internal logic and narrative flow of the manuscript. We reorganized early sections to progress more coherently from global pathophysiology → visceral adiposity → EAT biology → diagnostic limitations → novel biomarkers. Transitional sentences have been added to facilitate continuity. Page 3, lines 96-98; page 3, lines 119-121; page 4, lines 132-133; page 5, lines 193-195.

Third, in response to the reviewer’s observation about the focus and purpose of the review, we added a new paragraph in the Section 1.1 (“Aim of the review”), where we clearly state the translational objective of the article: a narrative review centered on EAT as an upstream mechanistic factor in diabetic HFpEF, integrating both EAT-derived adipokines and systemic biomarkers that complement EAT-related mechanisms. We also added a brief Methods-of-Review section describing the search strategy. Page 2, lines 42-60

Fourth, we made a minor modification to the manuscript title to better reflect the translational and integrative nature of the review, while maintaining EAT as the central mechanistic axis.

Finally, several structural improvements were implemented: splitting Chapter 3 into plasma biomarkers and imaging, adding transition paragraphs, incorporating a new HFpEF endotype table (Table 2), expanding the echocardiography subsection, modifying Table 1, adding comments on analytical/biological confounders, and expanding Conclusions with a new figure and practical pathway.

All changes are highlighted in yellow in the revised manuscript.

We believe these revisions improve clarity, reinforce the central role of EAT, and make the manuscript more coherent and aligned with the review’s stated aims.

Reviewer 2 Report

Comments and Suggestions for Authors
  1. Please, clearly declare the paper as a narrative review with translational aims and avoid any language that suggests causal effect. Add in the Introduction a paragraph that explains evidence levels and the limits of observational data. This change will align expectations with your own call for harmonization.
  2. Add a new methods-of-review section that describes how you selected studies (databases, date range, inclusion/exclusion, risk-of-bias approach). Even for a perspective, basic transparency will reduce selection bias concern.
  3. Please expand the imaging section with a minimal core protocol for echo-EAT: anatomical window, cardiac phase, caliper placement, and a recommended training set. Also propose an interim reporting template and an MCID hypothesis to guide serial monitoring. If you cannot provide consensus numbers, say this is a priority for future consensus.
  4. Upgrade Table 1 by adding, for each biomarker, the pre-analytical conditions, the assay platform examples, and whether the cut-off is assay-specific. Add footnotes about BMI and eGFR adjustments and propose stratified reference ranges. If you lack data, mark the cell as “unknown—needs validation” and declare as limitation.
  5. Temper claims in Table 2 by clarifying the role of weight loss as a mediator for EAT changes with GLP-1RA and lifestyle. Add a sentence that current signals are mechanistic and not equivalent to event reduction unless supported by outcome trials.
  6. Include a small, practical pathway figure for primary care: who to test (age, duration of T2D, BMI), which panel (sST2, Gal-3, GDF-15), how to repeat, and when to refer for imaging. Make clear that this is hypothetical and requires validation; add in the Conclusions that cost-effectiveness and equity analysis are still missing and must be done before broad adoption.
  7. Add a brief paragraph on potential confounders that are not controlled in cited cohorts: obesity class, renal function, liver fibrosis, sleep apnea, physical activity, and medication mix. Ask future studies to use multivariable models, propensity methods, or causal diagrams to reduce bias. If you cannot revise the synthesis, list this as a limitation.
  8. To increase translational value, propose a biomarker hierarchy with functional roles: sST2 as dynamic remodeling signal, Gal-3 as fibrosis residue marker, and GDF-15 as stress/adaptation index. State that this hierarchy is provisional and must be tested with decision-curve analysis and NRI/IDI in T2D cohorts. Place this at the end of the Conclusions as a research agenda.
  9. Consider adding a table that maps specific HFpEF endotypes (obese, hypertensive, atrial, right-sided pulmonary vascular) to expected EAT burden and to the biomarker triad. If this mapping is not yet evidence-based, clearly label it as hypothesis-generating and keep it as a limitation.
  10. In Figure 1, explain that the mechanisms shown (lipotoxicity, oxidative stress, fibrosis, calcium dysfunction) are conceptual and not proven causal pathways. Write this clearly in the legend or discussion.
  11. In Figure 2, clarify that the endocrine/paracrine actions of EAT are based on associative and experimental evidence, not clinical trials. Mention that direct causal validation is still missing.
  12. In Figure 3, add a short explanation of how to combine biomarkers with imaging and specify that this proposal is hypothetical and needs validation in prospective studies.
  13. In Table 1, include for each biomarker: the biological meaning, assay method, pre-analytical conditions, and whether the cut-off is assay-specific. Mention that obesity and kidney function can change these levels.
  14. In Table 2, clarify whether the EAT reduction observed with SGLT2i or GLP-1RA is independent of weight loss. If you cannot provide this data, you must write this as a limitation in the discussion.

Round 2

Reviewer 2 Report

Comments and Suggestions for Authors

I am satisfied that the authors have fully and appropriately addressed all concerns raised.